# Modelling the Distribution of the Red Macroalgae *Asparagopsis* to Support Sustainable Aquaculture Development

James O'Mahony [1,2] , Rubén de la Torre Cerro [1,3] and Paul Holloway [1,3,*]

1   Department of Geography, University College Cork, T12 K8AF Cork, Ireland; jamomah@gmail.com (J.O.); rubendltcerro@gmail.com (R.d.l.T.C.)
2   Centre for Marine & Renewable Energy Institute, University College Cork, P43 C573 Cork, Ireland
3   Environmental Research Institute, University College Cork, T23 XE10 Cork, Ireland
*   Correspondence: paul.holloway@ucc.ie

**Abstract:** Fermentative digestion by ruminant livestock is one of the main ways enteric methane enters the atmosphere, although recent studies have identified that including red macroalgae as a feed ingredient can drastically reduce methane produced by cattle. Here, we utilize ecological modelling to identify suitable sites for establishing aquaculture development to support sustainable agriculture and Sustainable Development Goals 1 and 2. We used species distributions models (SDMs) parameterized using an ensemble of multiple statistical and machine learning methods, accounting for novel methodological and ecological artefacts that arise from using such approaches on non-native and cultivated species. We predicted the current distribution of two *Asparagopsis* species to high accuracy around the coast of Ireland. The environmental drivers of each species differed depending on where the response data was sourced from (i.e., native vs. non-native), suggesting that the length of time *A. armata* has been present in Ireland may mean it has undergone a niche shift. Subsequently, researchers looking to adopt SDMs to support aquaculture development need to acknowledge emerging conceptual issues, and here we provide the code needed to implement such research, which should support efforts to effectively choose suitable sites for aquaculture development that account for the unique methodological steps identified in this research.

**Keywords:** machine learning; methane; mitigation; ruminant livestock; species distribution modelling





## 1. Introduction

Globally, agriculture contributes ~11% of total anthropogenic Green House Gas (GHG) emissions [1], with the production systems of cattle and sheep responsible for up to 18% of this [2]. Around 43% of GHG emissions are made up of enteric methane ($CH_4$) [3], which has a higher global warming potential than carbon dioxide ($CO_2$—approximately 28-fold [4]), meaning targets for reducing global warming will prove difficult if reductions in methane emissions are not as actively addressed as $CO_2$ emissions [5]. Fermentative digestion by ruminant livestock is one of the main ways enteric methane is produced as a by-product of anaerobic fermentation of organic feed matter [6,7]. Multiple challenges therefore exist to facilitate feeding an increasing global population in a more sustainable manner, with agriculture needing to identify methods to improve efficiencies [8], as well as aligning with Sustainable Development Goals (SDG) 1 and 2. Decreasing enteric $CH_4$ from ruminants consequently poses a unique opportunity to support resilience in response to climate change, with several strategies currently being explored to reduce $CH_4$ emissions [9–11].

Natural feed ingredients, notably those that sustainably decrease the environmental impact of food production, are increasingly becoming more important to consumers and producers [7]. The seaweed genus *Asparagopsis* is emerging as an active, innovative, and regenerative cleaner production feed for the wider agriculture sector [12]. Recent studies have shown that enteric $CH_4$ could be virtually eliminated using this genus as a feed

ingredient [7,13–15]. For example, Kinley et al. [7], investigated the effects of including *Asparagopsis* in feedlot beef cattle, demonstrating that when included in the high grain diet at 0.05%, 0.10%, and 0.20%, there was a decrease of $CH_4$ production of 9%, 38%, and 98%, respectively. Moreover, its inclusion enhanced growth rate in the steers, did not affect meat quality, and the anti-methanogenic compound bromoform was not detected in meat, fat, organs, or faeces of any of the steers.

This is particularly pertinent in the Republic of Ireland, where the agriculture sector is the most significant contributor to overall GHG emissions at 33.9% (~60.93 million tonnes carbon dioxide equivalent—$MtCO_2$ eq), with methane being the largest contributor of that figure at 64.5% [16]. When coupled with responses to recent fodder crises, the Republic of Ireland has seen an increase of dairy cow numbers by 27% and milk production by 40% in the last five years alone [16]. This increasing herd size can increase GHG emissions, notably, the release of significant amounts of $CH_4$. Subsequently, the value of *Asparagopsis* additions to cattle feed is increasingly being explored by policy experts, land managers, and government agencies, meaning research into identifying sustainable and suitable sites for aquaculture development is needed to support any climate change mitigation efforts.

Species distribution models (SDMs) are a powerful spatial analytical tool for studying the geographic distribution of a range of taxa [17,18], providing a methodological framework for researchers and practitioners to quantitatively assess the relationship between species distributions and environmental factors. SDMs project relationships in both environmental and geographic space using a variety of statistical methods and machine learning algorithms [19], and have been widely used for various applications, including aquaculture [20–22]. For example, Westmeijer et al. [22] used SDMs to assess the habitat suitability of nine temperate macroalgae species in Europe, identifying that temperature made the largest contribution to determining distributions, with the authors concluding that such analysis can support the selection of target species for seaweed aquaculture and support optimal growth conditions.

However, a challenge with identifying suitable cultivation sites for *Asparagopsis* is the complex and cryptic lineages of the different species, meaning any models developed could be compounded by uncertainty related to their native and non-native distributions [23]. A central assumption of SDMs is that species are in equilibrium with their environment, which may be violated for introduced or cultivated species [21]. During model parameterisation, it is therefore essential to the consider invasion (or cultivation) stage and the absence of equilibrium of the species in its new environment [24,25]. Moreover, distribution projections assume that species retain their niche [26], known as niche conservatism (i.e., where a species is only able to invade areas of similar ecological conditions that are found in their native range [27]). However, many species can shift their niche after introduction to a new environment. For example, the seaweed *Caulerpa taxifola* lives at different depths in its native and invaded areas [28,29]. The rapid evolution in several traits has been observed in many non-native species, making it possible for their fundamental niche to be modified [30].

Solutions to modelling the distribution of non-native species are not straightforward [25], but several methods can be adopted to overcome such limitations to model the distribution of non-native species more effectively and accurately [31]. For example, Verbruggen et al. [20] developed an SDM for the highly invasive species *Caulerpa cylindracea*, using training data from both the native and non-native range, identifying that at a global scale large parts of the coasts of Australia (native region) and the Mediterranean Sea (non-native region) had conditions suitable for macroecology, with the models for the non-native range predicting the species beyond the extension of the presently known range. To-date, little research has been conducted towards exploring whether the environmental and subsequent geographic distributions of *Asparagopsis* differ among their native and non-native ranges, which could have implications for any aquaculture developments stemming from such analytical models.

With the potential for the red algae genus *Asparagopsis* to be used in the mitigation of methane across a global agriculture sector, a vital part of any sustainable aquaculture process will be to identify suitable sites for its distribution for optimal growth, cultivation, and harvesting, with SDMs a primary analytical tool that can be used to achieve this. However, such models are complicated by challenges associated with niche conservatism and an absence of equilibrium in the non-native range, meaning research is needed to explore robust methods for parameterisation to inform agriculture. Here, we explore four main research questions with the overall aim of identifying suitable and sustainable cultivation sites for *Asparagopsis* for the island of Ireland: (1) What is the current suitable habitat for *Asparagopsis* spp.? (2) What are the most important environmental variables determining *Asparagopsis* spp. distributions? (3) Are *Asparagopsis* spp. in equilibrium in their non-native Irish range? and (4) What are the most suitable cultivation sites?

## 2. Materials and Methods

### 2.1. Study Species and Area

The genus *Asparagopsis* contains two accepted species taxonomically, *Asparagopsis armata* and *Asparagopsis taxiformis*. The species are morphologically and ecologically distinct: *A. armata* is an epiphyte attaching to other seaweed utilising barbs [32], while *A. taxiformis* is associated with sand-covered habitats, having a well-developed rhizomatous system for anchorage and lack barbs [33]. *A. armata* and *A. taxiformis* have a distinct geographical distribution and can overlap in some areas. *A. armata* is endemic to the southern hemisphere in cool-temperate waters. The species consists of two cryptic lineages, one where it is naturally distributed along western and southern Australia and New Zealand and the other in the north-east Atlantic and Mediterranean coasts where it is non-indigenous [23,34]. *A. taxiformis* is cosmopolitan in warm-temperate to tropical areas where it is widely distributed [35]. Within Europe, there are differences in the date of arrival of these species. *A. armata* is considered a Lessepsian immigrant, first reported in 1923 on the Algerian coast [36]. In 1925 it was recorded in France, then arriving in Ireland at Galway in 1941 [32]. It is now well established in open sandy pools of lower intertidal and subtidal zones, found on rocks or attached to other macroalgae (mainly *Ulva* spp.) [32]. In contrast, *A. taxiformis* is considered a pre-Lessepsian immigrant or native to the Mediterranean [37], since it was first recorded in the Mediterranean in 1813 in Egypt (Delile, 1813). According to the phylogeographic study by Andreakis et al. [37], the first lineage of *A. taxiformis* into the Mediterranean (Atlantic lineage 3) is confined to the south-east perhaps due to its low sensitivity to low winter seawater temperatures. The Indo-Pacific lineage 2 expanded the range of *A. taxiformis* into the north-eastern part of the Mediterranean during the second half of the 20th Century due to its lower minimum need of temperature for survival. The appearance of this lineage, especially on the south coast of Portugal, shows that the lineage has now been established in the north-eastern Atlantic [34]. *A. taxiformis*' fast dispersal rate and prolific vegetative reproduction, coupled with the effects of climate change, might result in expansion of its range into areas of the north-east Atlantic where it is currently not found [38,39].

Ireland is situated in the North Atlantic in north-west Europe (Figure 1), hosting a temperate maritime climate [40]. Large areas of the western, south-western, and northern coastlines are dominated by rocks, large bays, and estuaries, while the eastern and south-eastern coastlines are low-lying and soft sedimentary areas [41–43]. In 1996, a commercial cultivation farm for *A. armata* was set up in Ard Bay, Co. Galway. This site was chosen because *A. armata* had been found there since 1941. In 1998, a 1 ha farm was constructed and first cultivation trials took place. On this site, *A. armata* was cultivated using vegetative propagation of the gametophyte, where it is cultivated by connecting it to rope made of discarded twisted monofilament netting [44].

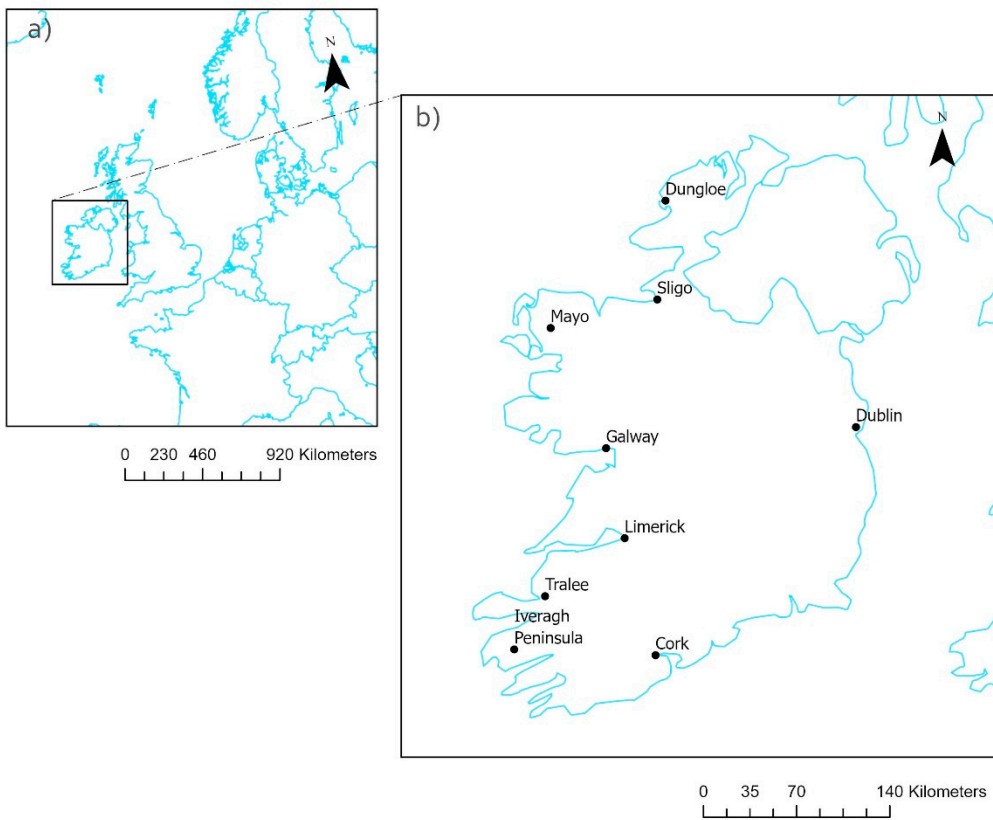

**Figure 1.** Map of the study sites (**a**) the Republic of Ireland situated within Ireland and (**b**) The Republic of Ireland with locations mentioned in the text documented.

*2.2. Data Collection*

Presence data for *A. armata* and *A. taxiformis* were obtained from the Global Biodiversity Information Facility using the *rgbif* package [45] in R studio [46]. Following data cleaning, 53 records of *A. armata* were obtained in the non-native Irish range and 103 records in the native New Zealand range [47]. Similarly, 75 records of *A. taxiformis* were obtained in the non-native Portuguese (Azores and Madeira) range and 699 records in the native Australian range [48]. Environmental data were obtained from Bio-ORACLE [49,50] and MARSPEC [51] databases using the *sdmpredictors* package [52]. All variables had a spatial resolution of 5 arc minutes, approximately 9 km. To avoid any potential multicollinearity problems, a Principal Component Analysis (PCA) was initially carried out on all variables to visualize the correlation between the environmental variables and identify the main environmental gradients in the region to be used in the modelling process. Using a PCA, environmental variables that were not collineated and significantly contributed to the overall environmental variation were selected. The *ade4* package [53] was used to perform this pre-analysis, and these results are presented in Supplementary Information 1. The variables used in subsequent data analysis are presented in Table 1, along with ecological justification for their inclusion.

**Table 1.** Information on the geospatial environmental layers used in the modelling framework, along with justification for their inclusion.

| Layer | Justification |
|---|---|
| Mean of diffuse attenuation | Diffuse attenuation, which is an indicator of light availability [54]; this light availability is important as it drives photosynthesis and growth of seaweeds [55]. |
| Dissolved oxygen | Significant contributor in PCA analysis (Supplementary Information 1). |
| Nitrate | The nutrient Nitrogen limits seaweed growth [55]. |
| pH | Significant contributor in PCA analysis (Supplementary Information 1). |
| Phosphate | The nutrient phosphorous limits seaweed growth (Roleda and Hurd, 2019). |
| Sea surface temperature range | Temperature is a primary range limiting factor [33]. |
| Temperature of warmest month | Temperature is a primary range limiting factor [33]. |
| Mean sea surface salinity | Significant contributor in PCA analysis (Supplementary Information 1). |
| Distance from shore | Distance to shore as *A.armata* is mainly found in the sublittoral zone [44]. |
| Bathymetry | Bathymetry as *A.armata* is mainly found in the sublittoral zone [44]. |
| *Ulva lactuca* species distribution | *A.armata* is an epiphyte that attaches to other seaweeds utilising its barbs [32]. |

### 2.3. Data Analysis

All code is presented in Supplementary Information 2. Two models for each species were built, one that accounted for presence data from the non-native range only and one that accounted for presence data from both the native and non-native ranges. As no absence data were available for the species, 10,000 pseudo-absences were randomly drawn (following [31]). For models using data from only the non-native range, pseudo-absences were also drawn only from the non-native range, whereas for models parameterised on both the native and non-native occurrence data, 5000 pseudo-absences were drawn from both the native and non-native ranges, totalling 10,000, with equal weighting given regardless of location. Table 2 summarises these different models. The choice of pseudo-absence has been found to impact the results of SDM projections [31,56], so we decided to replicate data analysis three times with different pseudo-absences selections.

**Table 2.** Information on the models with source of the response data and pseudo-absence data.

| Model | Presence | Pseudo-Absence |
|---|---|---|
| *A. armata* (non-native only) | Ireland | Ireland (10,000) |
| *A. armata* (native and non-native) | Ireland<br>New Zealand | Ireland (5000)<br>New Zealand (5000) |
| *A. taxiformis* (non-native only) | Portugal | Portugal (10,000) |
| *A. taxiformis* (native and non-native) | Portugal<br>Australia | Portugal (5000)<br>Australia (5000) |

PCA was again undertaken for the variables presented in Table 1 prior to fitting the analytical models. We implemented two models for *A. armata*, one that included a proxy for biotic interactions and one that did not. The former would allow for identification of sites that could be harvested naturally, while the latter would allow for identification of sites for vegetative propagation of the gametophyte. The species distribution of *A armata* and *A. taxiformis* were modelled by running six different SDM methodologies implemented within the *biomod2* package [57]. The six SDM methodologies included one regression method: Generalized Linear Model (GLM); two classification methods: Classification Tree Analyses (CTA) and Flexible Discriminant Analysis (FDA); and three machine learning

methods: Generalized Boosting Model (GBM), Random Forest (RF) and Artificial Neural Networks (ANN). Studies have shown that different modelling techniques can produce different results for the same species and datasets [58,59], meaning to obtain a consensus distribution, an ensemble forecast distribution was calculated as the average of all distributions across all modelling techniques and pseudo-absences replicates. The predictions from individual models were ensembled in four ways: the mean, median, confidence interval (upper) and confidence interval (lower) of habitat suitability.

Models were evaluated after parameterisation to justify the acceptance of projections for their intended purpose [60], in this case identifying suitable cultivation sites. We performed 3-fold cross-validation of the data by splitting the occurrence records 70:30 into training and testing data following best practice in the species distribution literature when independent test data are not available [18,61,62]. The model's discriminatory power between presence and absence was assessed using two different metrics to provide an accurate assessment of the models. The area under the curve (AUC) statistic of a receiver operating characteristic (ROC) [63] is a threshold-independent metric and widely used in SDM research [17]. It has a value range between 0 and 1, with a value below 0.5 deemed no better than a random selection and a value of 1 representing a highly accurate model. True Skill Statistic (TSS) measures the performance of models generating presence-absence predictions, with values of 0 indicating no agreement and 1 indicating perfect agreement [64].

## 3. Results

We identified large geographic variations in *Asparagopsis* spp. distributions within the coastal zone of Ireland (Figures 2 and 3; Table 3; Supplementary Information 3). When *A. armata* models were parameterised using the non-native range only and a proxy for biotic interactions (Figure 2a–d), we identified medium (>0.5) to high (>0.75) habitat suitability along the west coast of Ireland from Dungloe (co. Donegal) in the north to Cork in the south; however, when models were parameterised using both the native and non-native ranges (Figure 2e–h), the suitable environmental conditions were predicted in much smaller areas. Areas of medium (>0.5) to high (>0.75) suitability were still projected predominantly in the west of the country, mainly near Galway and in north west Kerry near Tralee, as well as near the Iveragh peninsula in south west Kerry and Cork. When modelled without biotic proxies (Supplementary Information 3), distributions of *A. armata* extended further out from the coast, suggesting there are suitable abiotic conditions for aquaculture through vegetative propagation of the gametophyte. No locations were projected as suitable for *A. taxiformis* using only the non-native Portuguese range as training data (Figure 3a–d), and only the median ensemble model using both native and non-native occurrence data (Figure 3e–h) identified two small areas with low habitat suitability (>0.25) in Galway Bay and the Shannon estuary near Limerick. Such results suggest that the absence of *A. taxiformis* from Irish waters is most likely a factor of limiting environmental conditions rather than dispersal ability.

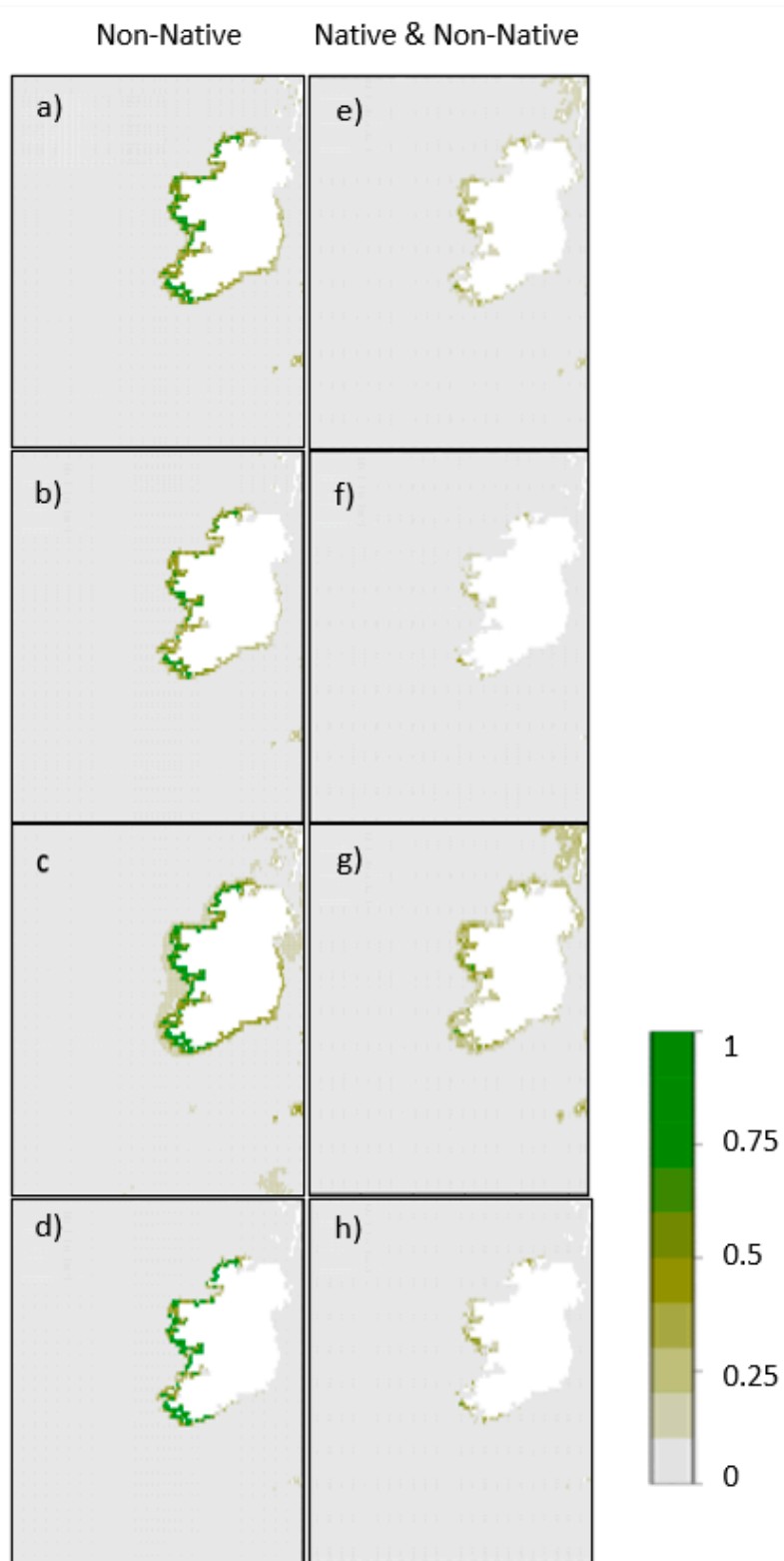

**Figure 2.** Plot showing the geographic projection of *Asparagopsis armata* parameterized using non-native data only represented as (**a**) the mean of ensembled habitat suitability across all iterations, (**b**) the median, (**c**) the upper confidence interval and (**d**) the lower confidence interval, as well as *Asparagopsis armata* parameterized using native and non-native data represented as (**e**) the mean of ensembled habitat suitability across all iterations, (**f**) the median, (**g**) the upper confidence interval and (**h**) the lower confidence interval.

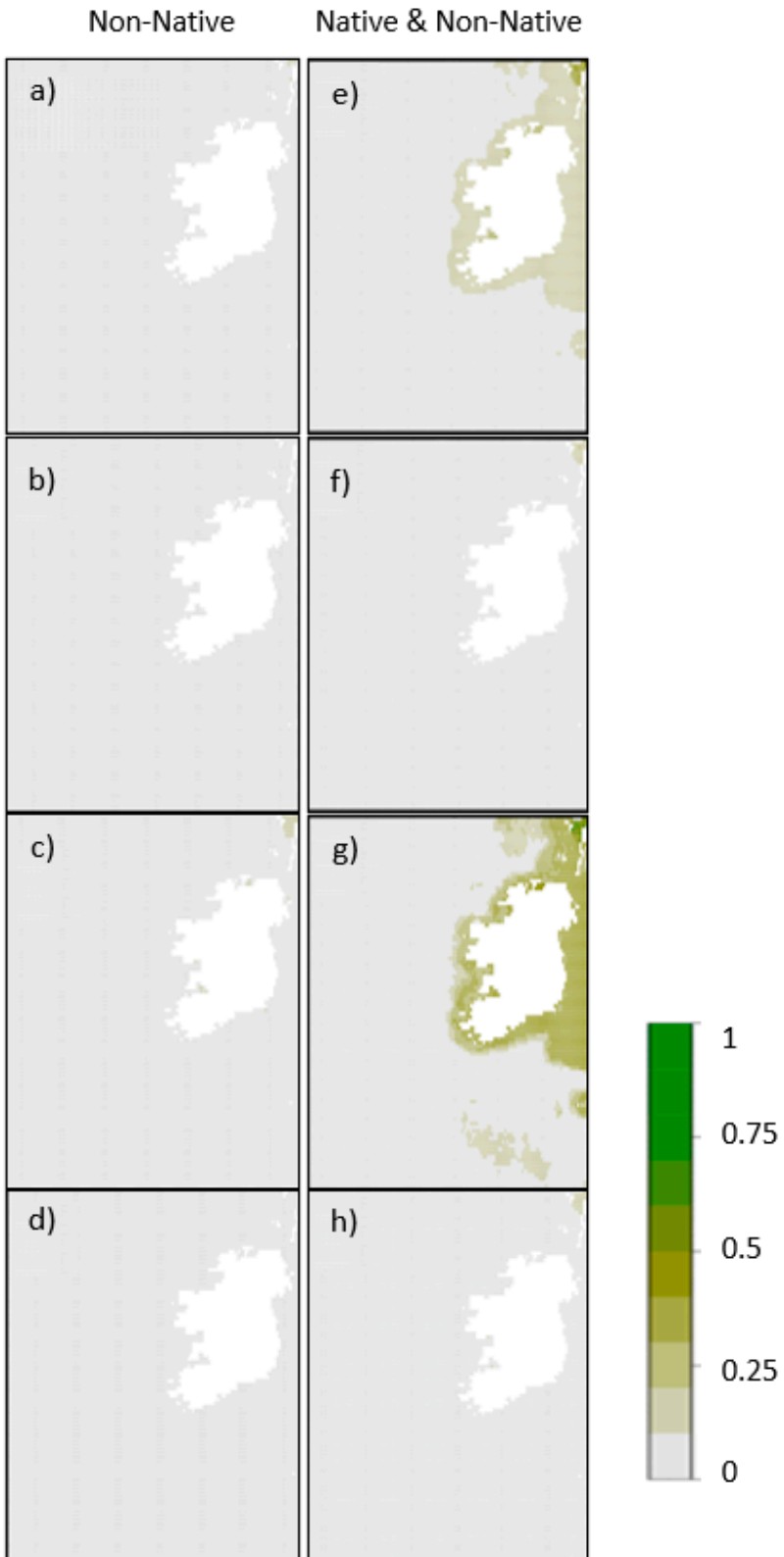

**Figure 3.** Plot showing the geographic projection of *Asparagopsis taxiformis* parameterized using non-native data only represented as (**a**) the mean of ensembled habitat suitability across all iterations, (**b**) the median, (**c**) the upper confidence interval and (**d**) the lower confidence interval, as well as *Asparagopsis taxiformis* parameterized using native and non-native data represented as (**e**) the mean of ensembled habitat suitability across all iterations, (**f**) the median, (**g**) the upper confidence interval and (**h**) the lower confidence interval.

**Table 3.** Accuracy metrics for the different models. CI = confidence interval. AUC = area under the curve. TSS = true skill statistics.

| Model | Metric | Mean Ensemble | CI (Lower) Ensemble | CI (Upper) Ensemble | Median Ensemble |
|---|---|---|---|---|---|
| *A. armata* (non-native) | AUC | 0.999 | 0.999 | 0.999 | 0.998 |
| *A. armata* (non-native) | TSS | 0.985 | 0.985 | 0.984 | 0.984 |
| *A. armata* (non-native and native) | AUC | 0.998 | 0.994 | 0.998 | 0.995 |
| *A. armata* (non-native and native) | TSS | 0.968 | 0.961 | 0.971 | 0.957 |
| *A. taxiformis* (non-native) | AUC | 1.000 | 1.000 | 1.000 | 0.999 |
| *A. taxiformis* (non-native) | TSS | 0.994 | 0.994 | 0.994 | 0.994 |
| *A. taxiformis* (non-native and native) | AUC | 0.998 | 0.979 | 0.998 | 0.994 |
| *A. taxiformis* (non-native and native) | TSS | 0.978 | 0.947 | 0.975 | 0.918 |

This is supported when we consider the importance of the environmental variables to distribution projections (Table 4). PCA identified a range of environmental variables to include in the different combinations of our final models (Supplementary Information 4). Temperature was important in determining the distribution of *A. taxiformis*, with temperature of the warmest month ($0.6 \pm 0.06$) the second highest variable when only the non-native range was considered, after nitrate ($0.74 \pm 0.09$). However, when *A. taxiformis* was parameterised with occurrences from both the native and non-native range, mean diffuse attenuation ($0.47 \pm 0.05$), nitrate ($0.30 \pm 0.06$) and pH ($0.20 \pm 0.05$) were considered the most important variables in determining the distribution, with temperature ($0.14 \pm 0.01$) less important but still retained in final models after PCA. For *A. armata*, the distribution of *Ulva lactuca* ($0.61 \pm 0.09$) was the most important variable in determining habitat suitability, followed by distance from the shore ($0.2 \pm 0.08$) and bathymetry ($0.2 \pm 0.07$) when models parameterised only on the non-native occurrence data were utilised. When both native and non-native occurrences were used, distance from shore ($0.62 \pm 0.01$), bathymetry ($0.33 \pm 0.04$) and *U. lactuca* distribution ($0.19 \pm 0.04$) remained important. The varying importance of environmental variables to the final model suggests that either species in the non-native ranges are not yet at equilibrium with the environment or that some niche shift has occurred.

**Table 4.** Variable impact on habitat suitability for the *Asparagopsis* spp. Values refer to mean and standard deviation across all cross validation runs and difference pseudo-absence combinations. No values are returned when variables were not included in final model parameterisation. SST = sea surface temperature.

| Environmental Variables | *A. armata* (Non-Native Only) | *A. armata* (Native and Non-Native) | *A. taxiformis* (Non-Native Only) | *A. taxiformis* (Native and Non-Native) |
|---|---|---|---|---|
| Mean of diffuse attenuation | 0.08(±0.02) | 0.16(±0.04) | 0.59(±0.06) | 0.47(±0.05) |
| Dissolved oxygen | 0.08(±0.02) | 0.10(±0.05) | | |
| Nitrate | 0.05(±0.02) | 0.08(±0.04) | 0.74(±0.09) | 0.30(±0.06) |
| pH | 0.09(±0.05) | 0.07(±0.03) | 0.21(±0.06) | 0.20(±0.05) |
| Phosphate | 0.1(±0.06) | 0.13(±0.06) | | |
| SST range | 0.1(±0.05) | 0.03(±0.01) | 0.33(±0.03) | 0.07(±0.02) |
| Temperature of warmest month | 0.06(±0.05) | 0.06(±0.02) | 0.60(±0.06) | 0.14(±0.01) |
| Mean sea surface salinity | 0.1(±0.03) | 0.07(±0.02) | 0.43(±0.06) | 0.06(±0.06) |
| Distance from shore | 0.2(±0.07) | 0.62(±0.01) | | |
| Bathymetry | 0.2(±0.08) | 0.33(±0.04) | | |
| *Ulva lactuca* species distribution | 0.61(±0.09) | 0.19(±0.04) | | |

## 4. Discussion

The overarching aim of this research was to assess the ability of analytical models to identify current suitable habitat for *Asparagopsis* spp. in Ireland for potential cultivation to support mitigation efforts at reducing enteric $CH_4$. Through this process, we explored how considerations, such as species equilibrium and non-native ranges, impacted results from an ecological and methodological perspective. Results indicated a large area of suitable habitat for *A. armata*, across all model iterations (Figure 2), but results for *A. taxiformis* were more restricted (Figure 3). When the model for *A. armata* was built with occurrence data from both native and non-native ranges, smaller areas of suitable habitats were identified (Figure 2), suggesting that this species is not yet at equilibrium with its environment. Moreover, when modelled without a proxy for biotic interactions, we identified a larger suitable area for *A. armata* suggesting the potential for vegetative propagation of the gametophyte would be suitable (Supplementary Information 3). The impact of methodological decisions on the overall results had a substantial impact (Figures 2 and 3, Tables 3 and 4), meaning the results and methodology of this study should be of interest to parties involved in mapping potential sites to support aquaculture development.

The mean of diffuse attenuation was one of the most important variables in determining habitat suitability for *A. taxiformis* (Table 4), with variables associated with temperature (e.g., SST range and temperature of the warmest month) also being important. This confirms the general recognition that temperature is the primary abiotic condition that shapes the geographic boundaries of seaweeds [65,66] and corroborates the findings of Guiry and Dawes [33] identifying temperature as a primary range limiting factor of *A. armata* distributions. However, we found that the distribution of *U. lactuca* was the most important variable in determining the habitat suitability of *A. armata* (Table 4). This biotic interaction of facilitation is a positive interaction for *A. armata* as it is an epiphyte that attaches to other seaweeds utilising its barbs [32]. This result supports the recent findings by Kraan and Barrington [44], where they identified *A. armata* growing on *U. lactuca*. However, when the model was parameterised with native and non-native occurrences, distance to shore was the most important variable in determining habitat suitability not *U. lactuca*. We believe that distance to shore is acting as a proxy for biotic interactions with other potential facilitator species within its native range, as closer to the shore there is a higher abundance of fast-growing kelp and other macroalgae to attach to [44]. The importance of incorpo-

rating biotic interactions within SDM is well-established [67–69]; however, abiotic factors often supersede biotic interactions within SDM due to their influence at a broader spatial scale [70,71]. Our results suggest there is a need to incorporate biotic interactions when determining cultivation sites, particularly as vegetative propagation of the gametophyte is needed, which could be received from wild populations; however, biotic interactions should be parameterised very differently depending on the source of the response data (i.e., native vs. non-native ranges).

To our knowledge, this is the first instance of *Asparagopsis* spp. being modelled using SDM. We found large areas of potentially suitable habitat (Figures 1 and 2); however, as these species are non-native to the study area, they may not be in equilibrium with the environment and subsequently violating key algorithm and modelling assumptions. The incorporation of occurrence records from both the native and non-native range simultaneously to build the model is one possible solution [25]. This process incorporates records that are likely to be in equilibrium with the environment in the native range while also including samples from the non-native range, which provides information about the expansion of the realised niche and the non-native area, which may provide valuable information about the species tolerance to climatic conditions that may not be present in the native range [21]. When both native and non-native data were used, projections were smaller for *A. armata* but larger for *A. taxiformis.* The rule of parsimony suggests that for habitat suitability models, a good projection will predict a potential area that is as small as possible [72], suggesting that the incorporation of native data in projecting *A. armata* was central to an effective modelling procedure. Subsequently, we identified three potential cultivation sites for *A. armata,* off the west coast of Galway, the north-west Kerry coast near Tralee and the south-west coast of the Iveragh peninsula (Figure 2). One of these smaller areas identified is the area near Ard Bay in Galway, which was identified by Kraan and Barrington [44] as a possible source pool for *A. armata* in Ireland, corroborating their study that highlighted the role the seaweed aquaculture facility may play, acting as a possible local source pool maintaining the gametophytic populations on the west coast of vegetative reproduction.

We noted the opposite relationship with *A. taxiformis,* with a larger predicted range when both non-native and native data were used (Figure 3). Currently, there are no occurrences of this species in Ireland, meaning we chose occurrences from Portugal because the occurrences found there have a lower minimum need of temperature and are the only occurrences that have been established in the north-eastern Atlantic [34]. There is an argument that *A. taxiformis* is considered either a pre-Lessepsian immigrant or native to the Mediterranean [37], so therefore occurrences from a southern hemisphere lineage were used where it may be native. When the model was built for *A. taxiformis* with occurrences from Portugal and Australia, the results for this model show that there are small areas of low habitat suitability identified in the Galway Bay and the Shannon estuary area near Limerick. When coupled with the abiotic drivers presented in Table 4, we posit that such results suggest that the absence of *A. taxiformis* from Irish waters is most likely a factor of limiting environmental conditions rather than dispersal ability.

All the models for *A. armata*, when accounting for the non-native data, had a broader predicted range compared to using both native and non-native data (Figure 2). This could suggest a niche shift. The rapid evolution in several traits has been observed in introduced species, which makes it possible for their fundamental niche to be modified [30]. The possible shift of niche can be seen when looking at the importance of environmental variables for determining habitat suitability (Table 4). For example, for the models looking at possible cultivation areas, those fit with non-native data infer that *Ulva lactuca*, pH, diffuse attenuation and dissolved oxygen were the most important in determining habitat suitability compared to the model fit with native and non-native data, where the distance to the shore, bathymetry, diffuse attenuation and phosphate were more important. Our findings suggest that future research is warranted to explicitly test the niche conservatism and shifts in this species, particularly as *A. armata* has been present in Irish waters for almost a century.

## 5. Conclusions

It is estimated that if just 10% of global ruminant producers adopted *Asparagopsis* as an additive to feed their livestock, it would have the same impact for the climate as removing 50 million cars from the world's roads [14]. Therefore, there is a pressing need to investigate new methods and analytical tools to support agriculture in line with carbon mitigation strategies, as well as supporting SDGs 1 and 2 to support feeding a growing population. Here, we utilised SDM to assess the habitat suitability of *Asparagopsis* spp. We parameterised models using both data from the native and non-native ranges to control for ecological artefacts that may occur when projecting species distributions in a non-native range. We found a large geographic area of suitable habitat for *A. armata* (Figure 2), but notably less for *A. taxiformis* (Figure 3). For all species, our models had good validation statistics (Table 3), but the environmental drivers of each species differed depending on where the response data was sourced from (i.e., native v non-native—Table 4). This suggests that the length of time *A. armata* has been present in Ireland may mean it has undergone a niche shift. Finally, we provide all the source code needed to undertake such research, which should support efforts to effectively choose suitable sites for aquaculture that account for the unique methodological steps identified in this research.

**Supplementary Materials:** The following are available online at https://www.mdpi.com/article/10.3390/agriengineering3020017/s1, SI1: Information and results of initial principal components analysis. SI2: Code to run the models. SI3: Results for *A. armata* without a biotic proxy. SI4: Results from principal components analysis. Additional references in SI [73,74].

**Author Contributions:** Conceptualization, J.O. and P.H.; methodology, J.O. and P.H.; formal analysis, J.O.; writing—original draft preparation, J.O., R.d.l.T.C. and P.H.; writing—review and editing, J.O., R.d.l.T.C. and P.H. All authors have read and agreed to the published version of the manuscript.

**Funding:** This research received no external funding.

**Institutional Review Board Statement:** Not applicable.

**Informed Consent Statement:** Not applicable.

**Data Availability Statement:** All data used are currently in open data repositories. Please see methods and SI2 for details.

**Acknowledgments:** We would like to thank the reviewers and editors for their constuctive comments.

**Conflicts of Interest:** The authors declare no conflict of interest.

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
