# Peer review of "Modelling the Distribution of the Red Macroalgae Asparagopsis to Support Sustainable Aquaculture Development"

_agriengineering, doi:10.3390/agriengineering3020017_

Round 1

Reviewer 1 Report

The article is very well written - it correctly introduces the topic, presents the material, methods, and then the obtained results, which are very interesting by the way. The conclusions are fair and the presentation is logical and clear. I believe that the article can be published as it is.

Author Response

Thank you very much for your kind words and positive review. 

Reviewer 2 Report

The article is well structured but some information seem to be repetitive and redundant. Some phrases must be shortened, some of them being too long and difficult to be understood. 

Author Response

Thank you for your review. Given the lack of specificity, we have had to use our own interpretation of what parts of the manuscript you found repetitive and redundant. We have concluded that the work looking at future projections is one angle of this research that was not central to the research, and perhaps resulted in a lot of repetition. As such, we have removed this section, which should streamline the article and make it read more clearly and concisely. We have also edited the document thoroughly to ensure no repetition or redundancy exists. 

Reviewer 3 Report

The manuscript shows an interesting study in which the main goal was to assess the ability of analytical models to identify current and future suitable habitats for Asparagopsis spp in Ireland for potential cultivation. To achieve this goal, the SDMs (Species distribution models) tool for spatial analysis was used, as it is commonly used for projecting the potential future range changes of species. The obtained projections justify the need for future research, as more macroalgae may be available locally to be used in fodders needed in a small scale industry, resulting in a reduction of methane emissions in the agricultural sector. For these reasons, I find this manuscript valuable and interesting, but nevertheless have a few minor comments on it.

Figure 2.: In the part of the caption on the geographic projection of Asparagopsis aramata parametrized with the native and non-native data, instead of a), b), c), there should be d), e), f), to correspond to the Figure 2.

Figure 2. a-d: in the caption above the figures shouldn't it be "non-native"? This question also applies to the caption under the figure.

Figure 3. In the part of the caption on the geographic projection of Asparagopsis taxiformis parametrized with the native and non-native data, instead of a), b), c), there should be d), e), f), to correspond to the Figure 3.

Line 414: was the pH in this case really the more significant variable?

Line 264: instead of “mean diffuse attenuation”, it should be “nitrate” (compare the text to Table 4).

Supplementary Information 1

Line 111 please remove the „Sea surface temperature (range)”, as this information is already given in Line 91.

Figure SI1 b: illegible labeling of environmental variables.

Supplementary Information 3

Figure SI3.1 b and SI3.2 b: illegible labeling of environmental variables.

Author Response

***Author responses

The manuscript shows an interesting study in which the main goal was to assess the ability of analytical models to identify current and future suitable habitats for Asparagopsis spp in Ireland for potential cultivation. To achieve this goal, the SDMs (Species distribution models) tool for spatial analysis was used, as it is commonly used for projecting the potential future range changes of species. The obtained projections justify the need for future research, as more macroalgae may be available locally to be used in fodders needed in a small scale industry, resulting in a reduction of methane emissions in the agricultural sector. For these reasons, I find this manuscript valuable and interesting, but nevertheless have a few minor comments on it.

***Thank you for the constructive feedback and positive comments on our research

Figure 2.: In the part of the caption on the geographic projection of Asparagopsis aramata parametrized with the native and non-native data, instead of a), b), c), there should be d), e), f), to correspond to the Figure 2.

***Thanks for spotting this typo. This has now been fixed.

Figure 2. a-d: in the caption above the figures shouldn't it be "non-native"? This question also applies to the caption under the figure.

***Correct, thanks for spotting this typo. This has now been fixed.

Figure 3. In the part of the caption on the geographic projection of Asparagopsis taxiformis parametrized with the native and non-native data, instead of a), b), c), there should be d), e), f), to correspond to the Figure 3.

***Thanks for spotting. This has now been fixed.

Line 414: was the pH in this case really the more significant variable?

***It was not the most significant variable, but it was one of the most important. We have subsequently reworded this section to remove any confusion.

Line 264: instead of “mean diffuse attenuation”, it should be “nitrate” (compare the text to Table 4).

***Correct. This was an oversight on our part. We have now fixed this. Thanks again for spotting.

Supplementary Information 1

Line 111 please remove the „Sea surface temperature (range)”, as this information is already given in Line 91.

***We have removed the duplicated entry

Figure SI1 b: illegible labeling of environmental variables.

***We have improved the figures and have made all labelling legible

Supplementary Information 3

Figure SI3.1 b and SI3.2 b: illegible labeling of environmental variables.

***We have improved the figures and have made all labeling legible